# Adapting Vision Transformers to Ultra-High Resolution Semantic Segmentation with Relay Tokens

**Yohann Perron**                                            *yohann.perron@enpc.fr*
*École française d'Extrême-Orient (EFEO)*
*Ecole Nationale des Ponts et Chaussées, IP Paris, Univ Gustave Eiffel, CNRS*

**Vladyslav Sydorov**                                     *vladyslav.sydorov@efeo.net*
*École française d'Extrême-Orient (EFEO)*

**Christophe Pottier**                                   *christophe.pottier@efeo.net*
*École française d'Extrême-Orient (EFEO)*

**Loic Landrieu**                                            *loic.landrieu@enpc.fr*
*Ecole Nationale des Ponts et Chaussées, IP Paris, Univ Gustave Eiffel, CNRS*

**Reviewed on OpenReview:** *https://openreview.net/forum?id=tidYprMlsg*

## Abstract

Current approaches for segmenting ultra-high-resolution images either slide a window, thereby discarding global context, or downsample and lose fine detail. We propose a simple yet effective method that brings explicit multi-scale reasoning to vision transformers, simultaneously preserving local details and global awareness. Concretely, we process each image in parallel at a *local* scale (high-resolution, small crops) and a *global* scale (low-resolution, large crops), and aggregate and propagate features between the two branches with a small set of learnable *relay tokens*. The design plugs directly into standard transformer backbones (*e.g.* ViT and Swin) and adds fewer than 2 % parameters. Extensive experiments on three ultra-high-resolution segmentation benchmarks, Archaeoscape, URUR, and Gleason, and on the conventional Cityscapes dataset show consistent gains, with up to 15 % relative mIoU improvement. Code and pretrained models are available at `https://archaeoscape.ai/work/relay-tokens/`.

## 1 Introduction

Most vision benchmarks operate on images of $224 \times 224$ pixels. In contrast, domains such as Earth observation and medical imaging routinely involve scenes spanning hundreds of megapixels—a regime we refer to as *ultra-high resolution* (UHR) (Sun et al., 2024; Ji et al., 2023; Guo et al., 2022; Shen et al., 2022; Chen et al., 2019). Segmenting such images requires both long-range reasoning to place objects within a global context and fine-grained spatial resolution, as targets of interest may occupy only a few pixels. This dual requirement makes it necessary to process extremely large images while preserving detailed local information. The quadratic token–token interactions in standard transformer layers (Dosovitskiy et al., 2021; Liu et al., 2021; 2022; Oquab et al., 2024) make off-the-shelf ViT-style architectures impractical for ultra-high-resolution semantic segmentation.

To circumvent this limitation, existing approaches either (i) apply sliding-window inference, which ignores long-range context, or (ii) downsample the input, which foregoes fine structures (see Fig. 1). Dedicated multi-scale architectures partly alleviate these issues, but at the cost of bespoke designs and full retraining (Chen et al., 2019; Wang et al., 2024; Guo et al., 2022; Ji et al., 2023). Recent linear-complexity transformers (Gupta et al., 2024; Zhu et al., 2024a) can process large images, yet still struggle to reconcile detail and context.

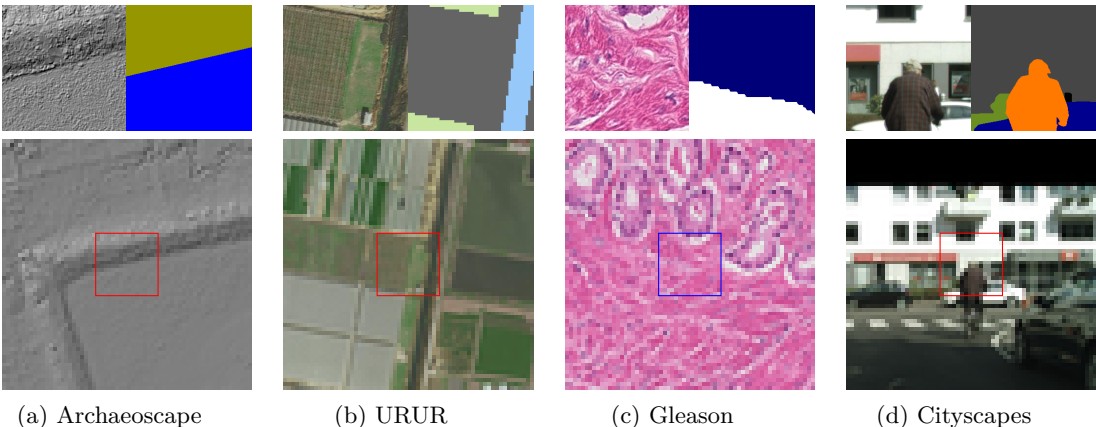

|  |  |  |  |
|---|---|---|---|
| (a) Archaeoscape | (b) URUR | (c) Gleason | (d) Cityscapes |

Figure 1: **Context and Details.** Examples of scenes where both the high-resolution details (top-left) and the low-resolution context (bottom) are necessary to correctly predict the true label maps (top right).

**Relay Tokens.** We introduce a plug-and-play mechanism that endows any ViT-style backbone with explicit multi-scale reasoning *without* altering its structure or discarding pretrained weights. Inspired by the Perceiver's latent arrays (Jaegle et al., 2021), we add a handful of learnable *relay tokens* that are shared between a local branch (high-resolution, small crops) and a global branch (low-resolution, large crops). During self-attention, these tokens collect features from one scale and propagate them into the other, effectively transmitting information across resolutions (Fig. 2). The resulting network increases parameter count by less than 2% (only 0.0005% when sharing patch embedding across scales) and incurs negligible runtime overhead compared to processing both local and global windows separately.

**Empirical Validation.** We evaluate our approach on three UHR segmentation benchmarks: aerial LiDAR (Archaeoscape), satellite photography (URUR), and histology slides (Gleason), as well as a conventional vision dataset (Cityscapes). We consider two widely used backbone families, ViT (Dosovitskiy et al., 2021) and Swin (Liu et al., 2021; 2022), as well as the specialised GLAM architecture (Themyr et al., 2023) and the linear-complexity Flatten-Swin (Han et al., 2023). Relay tokens consistently improve mIoU, by up to 15 %, while decreasing peak GPU memory by an order of magnitude relative to full-resolution processing.

Our contributions are:

- A simple, flexible, and drop-in modification of vision transformers that preserves pretrained weights and adds $\leq 2$ % parameters.
- Extensive experiments on UHR and classic vision benchmarks demonstrating substantial accuracy gains and memory savings compared to other scaling approaches.

## 2 Related Work

**Classical Semantic Segmentation.** Fully Convolutional Networks (Long et al., 2015) and U-Nets (Ronneberger et al., 2015) pioneered pixel-wise classification with deep neural networks. Subsequent architectures improved segmentation quality primarily through enhanced receptive fields, leveraging dilated convolutions (Chen et al., 2017a; 2018), pyramid pooling modules (Zhao et al., 2017), and boundary-focused feature extraction techniques (Ding et al., 2019; Takikawa et al., 2019; Chen et al., 2016). More recently, Vision Transformers (ViTs) (Dosovitskiy et al., 2021; Jain et al., 2023; Zhang et al., 2023; Kirillov et al., 2023) have demonstrated superior performance and versatility, shifting the segmentation landscape towards transformer-based architectures.

**Hierarchical Image Representation.** Segmenting high-resolution images poses substantial computational and memory challenges. Common approaches include sliding window techniques, which introduce boundary artefacts and lose global context, or downsampling preprocessing steps, which blur critical details (Tokunaga et al., 2019; Ho et al., 2021). To mitigate these limitations, multiple approaches have been proposed to

integrate local, high-resolution patches with broader low-resolution contexts at various network depths, ranging from late fusion (Tokunaga et al., 2019; Wang et al., 2024), intermediate bottleneck fusion (Schmitz et al., 2021), to progressive fusion schemes (Gu et al., 2018; Chen et al., 2019; Ho et al., 2021). Alternatively, spatial pyramids (Burt & Adelson, 1983; Lazebnik et al., 2006) explicitly generate multi-resolution feature representations, integrating them either at the final prediction stage (Guo et al., 2022) or incrementally throughout the network (Zhu et al., 2024b). Coarse-to-fine refinement methods iteratively improve segmentation predictions through a hierarchy of using increasingly detailed feature resolutions (Cheng et al., 2020; Shen et al., 2022). Our proposed relay token approach shares conceptual similarities with these hierarchical methods; however, unlike prior techniques, we retain pretrained transformer backbones unchanged and enable multi-scale reasoning through a small set of shared tokens.

**Efficient Architecture Backbone.** Another line of research rely on efficient architectures to directly handle large-scale inputs without explicit multi-scale fusion. Early solutions leverage the hierarchical structure inherent to convolutional networks (Chen et al., 2017a). More recent transformer variants reduce computational complexity with hierarchical window attention (Liu et al., 2021; Xie et al., 2021; Wang et al., 2021), clustering tokens spatially (Sun et al., 2024), or linear-complexity state-space models (Fu et al., 2024; Zhu et al., 2024a; Gupta et al., 2024). However, these methods typically require substantial architectural redesign and training from scratch or discard pretrained model parameters. In contrast, relay tokens require minimal modifications, retain pretrained weights, and seamlessly enable multi-scale reasoning capabilities into existing ViT-based architectures with minimal overhead.

**Token-Based Cross-Resolution Embeddings.** Several works combine multi-resolution information using latent vectors, referred to as *global tokens* (Themyr et al., 2023; Zhang et al., 2021) or *memory tokens* (Themyr et al., 2022). Unlike our relay tokens, these methods process the full-resolution, full-context image at every scale, limiting the effective context. Longformer (Zhang et al., 2021) and GLAM (Themyr et al., 2023) additionally introduce custom multi-resolution blocks on top of Swin (Liu et al., 2021), whereas relay tokens are a backbone-agnostic drop-in compatible with any ViT-style encoder. Moreover, their global tokens are local to each transformer block and are discarded afterwards; relay tokens persist throughout the network, propagating information across stages and input resolutions. FINE (Themyr et al., 2022) also employs persistent memory tokens and is not tied to a particular backbone, but its code has not been released, and its evaluation is limited to medical images.

## 3 Method

We propose a straightforward extension to Vision Transformers for efficient multi-scale image segmentation; see Fig. 2. Specifically, we segment large images (*e.g.*, $5000 \times 5000$) by simultaneously processing two windows: a small, high-resolution *local* window ($x^{\text{local}}$, *e.g.* $256 \times 256$) and a larger, downsampled *global* window ($x^{\text{global}}$, *e.g.* $1024 \times 1024$ downsampled to $256 \times 256$). These paired windows capture both high-resolution details and broader context. In Sec. 3.1, we describe the minimal changes required to adapt existing ViT models, and in Sec. 3.2, we detail our training losses.

### 3.1 Architecture

We first briefly review standard ViTs and then describe how to add relay tokens.

**Background: Vision Transformers.** A standard ViT (Dosovitskiy et al., 2021) segments images by first dividing them into fixed-size patches (typically $16 \times 16$ pixels). Each patch is then embedded to a $D$-dimensional vector by a projector $\phi^{\text{proj}}$, augmented with positional encodings, and processed through $B$ transformer blocks $\phi_1^{\text{block}}, \ldots, \phi_B^{\text{block}}$. A segmentation head $\phi^{\text{seg}}$ maps the resulting embeddings to pixel-level predictions. Formally, given the patches $p^{\text{local}}$ of the local window $x^{\text{local}}$, the segmentation predictions $z^{\text{local}}$ are defined as:

$$z^{\text{local}} = \phi^{\text{seg}} \circ \phi_B^{\text{block}} \circ \cdots \circ \phi_1^{\text{block}} \big( \phi^{\text{proj}}(p_i^{\text{local}}) + \text{pos}(p_i^{\text{local}}) \big), \tag{1}$$

where pos denotes the positional encoding function.

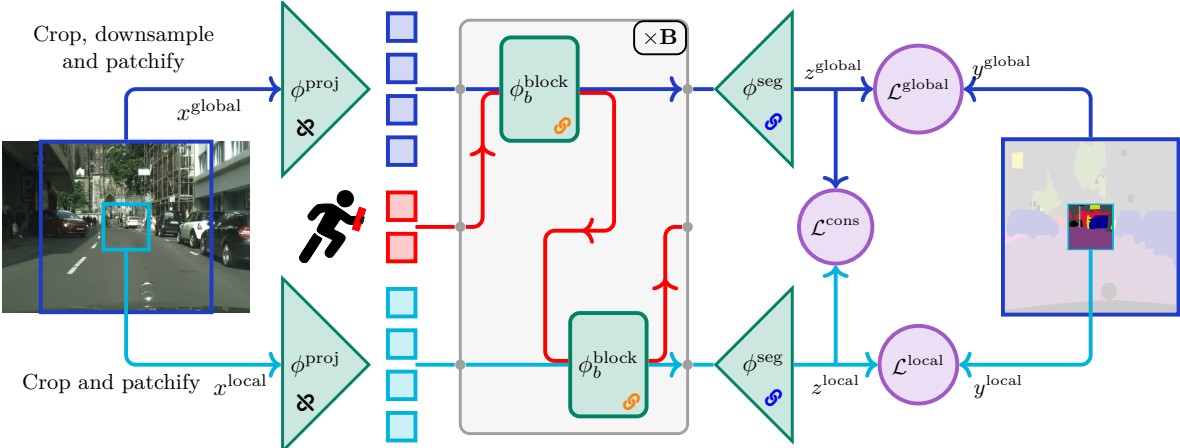

Figure 2: **ViT with Cross-Resolution Relay Tokens.** We simultaneously process a small high-resolution window $x^{\text{local}}$ and a larger low-resolution window $x^{\text{global}}$ *with a shared network* augmented with relay tokens ▢. In each of the $B$ consecutive transformer blocks, relay tokens are first added to the patch tokens of the global window and processed by the transformer. The updated relay tokens are then processed with the tokens of the local window and passed to the next block. The model applies supervision at each scale using losses $\mathcal{L}^{\text{global}}$ and $\mathcal{L}^{\text{local}}$, while enforcing consistency between resolutions via a cross-resolution loss $\mathcal{L}^{\text{cons}}$. We denote shared weights with 🔗 and independent parameters with ✂.

**Shared Weights.**   We extend this framework by jointly processing both local ($x^{\text{local}}$) and global ($x^{\text{global}}$) windows using the *same* transformer, with weights shared across all modules except the projection layers. We first embed the local and global patches $p^{\text{local}}$ and $p^{\text{global}}$ separately:

$$f_0^{\text{local}} = \phi_{\text{local}}^{\text{proj}}(p_i^{\text{local}}) + \text{pos} \tag{2}$$

$$f_0^{\text{global}} = \phi_{\text{global}}^{\text{proj}}(p_j^{\text{global}}) + \text{pos} \ , \tag{3}$$

**Relay Tokens.**   We add $R$ relay tokens $f_0^{\text{relay}} \in \mathbb{R}^{R \times D}$ learned as free parameters of the model. Each transformer block $b = 1, \ldots, B$ sequentially updates tokens in two steps:

$$\left[f_{b+\frac{1}{2}}^{\text{relay}}, f_{b+1}^{\text{global}}\right] = \phi_b^{\text{block}}\left(\left[f_b^{\text{relay}}, f_b^{\text{global}}\right]\right) \tag{4}$$

$$\left[f_{b+1}^{\text{relay}}, f_{b+1}^{\text{local}}\right] = \phi_b^{\text{block}}\left(\left[f_{b+\frac{1}{2}}^{\text{relay}}, f_b^{\text{local}}\right]\right) \ , \tag{5}$$

where $[\cdot, \cdot]$ concatenates tokens, and $b + \frac{1}{2}$ indicates the *intermediate* update of the relay tokens after processing the global tokens only. In Eqs. (4) and (5), the output of $\phi_b^{\text{block}}$ is split such that the first term corresponds to the $R$ updated relay tokens, and the second one the image token embeddings. After the final block, the relay tokens are discarded. We then apply the same segmentation head to the final local and global features:

$$z^{\text{local}} = \phi^{\text{seg}}(f_B^{\text{local}}) \tag{6}$$

$$z^{\text{global}} = \phi^{\text{seg}}(f_B^{\text{global}}) \ , \tag{7}$$

Algorithm 1 shows the code adding relays to a ViT.

Algorithm 1: **Vision Transformer With Relay Tokens.** PyTorch code to add relay tokens to a ViT, here '...' stands for standard ViT code. See Appendix for more details.

D – embedding dimension     R – number of relays

```python
class ViT_with_relays(nn.Module):
  def __init__(self, D, R, ...):
    ...
    self.relay = nn.Parameter(torch.randn(R, D))

  def forward(self, img_loc, img_glo):
    f_loc = patchify_and_project(img_loc)
    f_glo = patchify_and_project(img_glo)
    f_r = repeat(self.relay, batch_size)

    for block in self.blocks:
      f_r_glo = block(torch.cat((f_r, f_glo),1))
      f_glo, f_r = f_r_glo[:, R:], f_r_glo[:, :R]

      f_r_loc = block(torch.cat((f_r, f_loc),1))
      f_loc, f_r = f_r_loc[:, R:], f_r_loc[:, :R]
    ...
    return out
```

**Beyond ViTs.** Adding relay tokens to a vanilla ViT requires only a few extra lines of code (see Algorithm 1). Extending the same principle to hierarchical models like SwinV2 (Liu et al., 2022) or even more complex approaches such as GLAM (Themyr et al., 2023) is similarly simple: we copy the relay tokens for each window-attention block, then average their updated values across windows before applying the feed-forward layer.

## 3.2 Training

Our training procedure employs three complementary loss terms to encourage precise segmentation at both local and global scales, while enforcing consistency between them. All supervision leverages ground truth labels $y^{\text{local}}$ associated solely with the high-resolution local window.

**Local Supervision.** We supervise the local prediction $z^{\text{local}}$ using standard cross-entropy loss:

$$\mathcal{L}^{\text{local}} = \mathrm{H}\left(z^{\text{local}}, y^{\text{local}}\right) \ , \tag{8}$$

where $\mathrm{H}(z, y) = \sum_k y_k \log(z_k)$.

**Global Supervision.** We encourage the global-scale embeddings to focus primarily on context relevant to local-scale segmentation accuracy. We first crop the global predictions $z^{\text{global}}$ to the spatial extent of the local window, denoted as $\text{crop}(z^{\text{global}})$. Next, we apply average pooling (avg) to aggregate the high-resolution local labels $y^{\text{local}}$ into lower-resolution distributions. The global loss is:

$$\mathcal{L}^{\text{global}} = \mathrm{H}\left(\text{crop}(z^{\text{global}}), \text{avg}(y^{\text{local}})\right) \ . \tag{9}$$

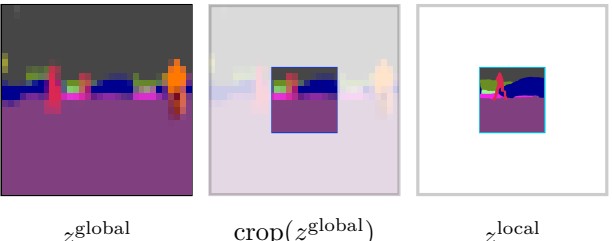

$z^{\text{global}}$      $\text{crop}(z^{\text{global}})$      $z^{\text{local}}$

Figure 3: **Predictions at Different Scales.**

**Consistency Loss.** Similarly to SGNet (Wang et al., 2024), we align predictions across scales with a consistency loss. Specifically, we match the cropped global predictions to the spatially averaged local predictions:

$$\mathcal{L}^{\text{cons}} = \mathrm{H}\left(\text{crop}(z^{\text{global}}), \text{avg}(z^{\text{local}})\right) \ . \tag{10}$$

This term encourages effective cross-resolution communication, which may reduce boundary discontinuities and enhance the segmentation consistency.

## 4 Experiments

We first present the datasets and baselines in Sec. 4.1. We then discuss our main experimental results (Sec. 4.2) and provide an extended analysis (Sec. 4.3), followed by an ablation study (Sec. 4.4).

### 4.1 Datasets and Baselines

We evaluate our method on four datasets that collectively cover LiDAR archaeology, satellite image analysis, histopathology, and urban-scene imagery. Performance is measured by mean Intersection-over-Union (mIoU).

**Datasets.** We evaluate all models on three ultra-high resolution image segmentation datasets and one computer vision dataset, illustrated in Fig. 4 (see Appendix for additional details):

- **Archaeoscape (Perron et al., 2024)** is a LiDAR–RGB dataset spanning $888\,\text{km}^2$ at 0.5m ground-sampling distance. It comprises 23 parcels of up to $720 \times 10^6$ pixels, densely annotated with four classes (water features, earthen mounds, temples, background). We report mIoU over the three foreground classes on the `test` split.

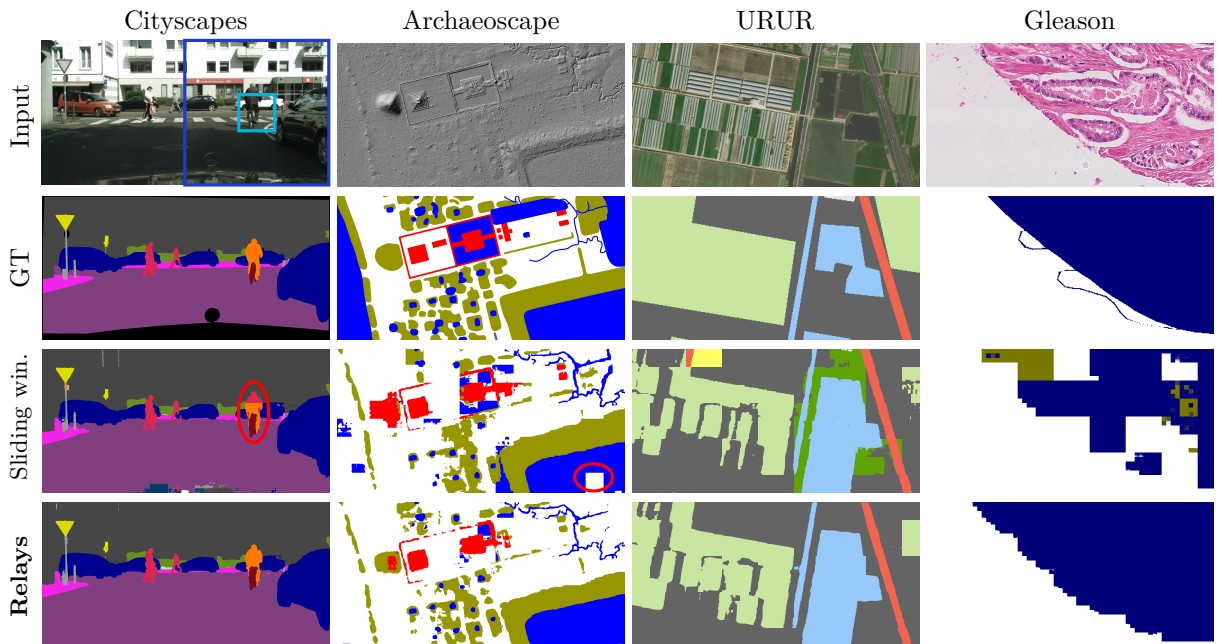

Figure 4: **Qualitative Results.** We visualise a $2048 \times 1024$ region from each dataset, comparing our multi-scale approach to the baseline sliding window. We highlight the extents of the **global** and **local** images and mark points of interest with ⭕.

- **URUR (Ji et al., 2023)** contains 3,008 satellite images of $5012 \times 5012$ pixels with eight land-cover classes. We follow the standard split (2,157 / 280 / 571 images for `train` / `val` / `test`).
- **Gleason (Karimi et al., 2020)** features 244 histology micro-array tiles ($5120 \times 5120$ pixels) across seven slides, labelled as benign or one of three Gleason grades. Official servers are no longer available to evaluate on the test set and recent works did not disclose their test split (Wang et al., 2024). We use slide 7 as our `test` set, train on the remaining slides, and reserve images with a *core number* under 15 for validation.
- **Cityscapes (Cordts et al., 2016)** offers 2,975 training and 500 validation street-view images at $1024 \times 2048$ resolution with 19-class dense annotations. *While not UHR*, this dataset provides a familiar benchmark to evaluate our design in a classic computer vision setting. We train on the `train` split and report mIoU on `val`.

**Baselines and Competing Methods.** We benchmark our approach against several approaches:

- **Sliding Window.** We evaluate three backbone ViT models — ViT (Dosovitskiy et al., 2021), SwinV2 (Liu et al., 2022), and GLAM (Themyr et al., 2023) in a sliding window setting, with and without relay tokens.
- **Scalable Vision Transformers.** We evaluate Flatten Swin — Swin configuration of Flatten Transformer (Han et al., 2023), which employs linear-attention modules inside a Transformer architecture. We also evaluate two ViT models able to scale to large windows thanks to their linear-complexity transformer replacement: xT (Gupta et al., 2024) and Vision Mamba (Zhu et al., 2024a).
- **Multi-Scale Approaches.** We evaluate three models that also process large images with a dual sliding window approach: GLNet (Chen et al., 2019), SGNet (Wang et al., 2024), and ISDNet (Gu et al., 2018). We use the window sizes recommended in the articles.
- **Classic Models.** We also evaluate classic models such as DeepLabv3 (Chen et al., 2017b), U-Net (Ronneberger et al., 2015), and PVTv2 (Wang et al., 2022) to provide context in terms of performance. To reflect the lesser memory usage compared to transformers, we use a larger sliding window size of 512 for CNN-based models.

Table 1: **Quantitative Evaluation.** Performance (mIoU) for different scaling strategies on three UHR and one vision dataset. We report the (+% mIOU) improvement of relay tokens over the sliding-window baseline.

| SW – sliding window | ↓ – subsampling | ⋆ – performance on the test set |
| MS – multi-scale | full – entire input image | † – performance on an undisclosed test set |

| Backbone | Scaling strategy | Archaeoscape | URUR | Gleason | Cityscapes (val) |
| | | up to 30K × 40K | 5012 × 5012 | 5120 × 5120 | 1024 × 2048 |
|---|---|---|---|---|---|
| ViT [1] | SW: 256 | 46.5 | 36.3 | 33.2 | 53.0 |
| + **relays (ours)** | MS: 256 + 1024 ↓ 4 | 46.7 (+0.2) | 40.4 (+4.1) | 49.1 (+15.9) | 61.0 (+8.0) |
| SwinV2 [2] | SW: 256 | 51.9 | 41.0 | 48.1 | 68.2 |
| + **relays (ours)** | MS: 256 + 1024 ↓ 4 | **57.8 (+5.9)** | 46.4 (+5.4) | 55.5 (+7.4) | 75.1 (+6.9) |
| Flatten Swin [3] | SW: 256 | 53.4 | 43.0 | 50.7 | 70.2 |
| + **relays (ours)** | MS: 256 + 1024 ↓ 4 | 55.2 (+1.8) | 45.8 (+2.8) | **57.7 (+7.0)** | 77.5 (+7.3) |
| GLAM [4] | SW: 256 | 52.5 | 43.9 | 47.7 | 72.9 |
| + **relays (ours)** | MS: 256 + 1024 ↓ 4 | 53.8 (+1.3) | 44.5 (+0.6) | 54.1 (+6.4) | 76.9 (+4.0) |
| xT [5] | SW: 1024 | 40.6 | 44.0 | 46.0 | 68.6 |
| Vision Mamba [6] | SW: 1024 | 45.9 | 31.6 | 22.1 | 44.8 |
| GLNet [7] | MS: 512 + full ↓ | - | 41.2 | - | 71.2 |
| SGNet [8] | MS: 512 + 1024 | - | - | 61.2 † | 70.4 |
| ISDnet [9] | MS: full + full ↓ 4 | 43.6 | 45.8 | 60.0 † | 76.0 ⋆ |
| DeepLabv3 [10] | SW: 512 | 49.7 | 41.7 | 46.3 | 56.6 |
| U-Net [11] | SW: 512 | 49.9 | 43.1 | 53.5 | 67.8 |
| PvTv2 [12] | SW: 256 | 52.1 | 41.2 | 48.6 | 66.1 |
| SOTA | | 52.1 [13] | **46.9 [14]** | | **87.4 [15]** |

[1] Dosovitskiy et al. (2021)  [2] Liu et al. (2022)  [3] Han et al. (2023)  [4] Themyr et al. (2023)  [5] Gupta et al. (2024)  [6] Zhu et al. (2024a)  [7] Chen et al. (2019)  [8] Wang et al. (2024)  [9] Guo et al. (2022)  [10] Chen et al. (2017b)  [11] Ronneberger et al. (2015)  [12] Wang et al. (2022)  [13] Perron et al. (2024)  [14] Ji et al. (2023)  [15] Erişen (2024)

**Notations.** We use the downsampling factor notation $n \downarrow k$ to indicate that an $n \times n$ image is downsampled $k$ times; *e.g.* $1024 \downarrow 4$ represents a $1024 \times 1024$ image downsampled to $256 \times 256$. The '+' operator denotes models combining multiple scales. For example, $256 + 1024 \downarrow 4$ refers to a model using a $256 \times 256$ local window and a $4\times$ downsampled $1024 \times 1024$ crop as global image. We refer to the entire input image as 'full'.

**Dataloader.** For each training step, we randomly select one pixel as the centre of both local and global crops. If more than 80% of the local window lies outside the main image, we skip that sample. We pad the out-of-bounds parts of the local and global windows with the image mean. We apply random scaling in $[0.5, 2]$ and random rotations in $[-90°, 90°]$, each with 0.5 probability. We do not use any test time augmentations.

**Implementation Details.** All models utilise the $S$ (small) variant of each backbone, initialised from ImageNet (Ridnik et al., 2021; Russakovsky et al., 2015), and four relay tokens. For all datasets, we train with the combined loss $\mathcal{L}^{\text{local}} + 0.1\,\mathcal{L}^{\text{global}} + 0.1\,\mathcal{L}^{\text{cons}}$ using `AdamW` (Kingma & Ba, 2015; Loshchilov & Hutter, 2019) with an initial learning rate of $10^{-4}$, no weight decay, a linear warmup for 5% of training, and a ReduceLROnPlateau (PyTorch 2.7) schedule based on validation accuracy. For URUR and Gleason with ViT and SwinV2 backbones, the patch embedding networks of both scales share parameters. In all other cases, including Cityscapes, Archaeoscape, and experiments with Flatten Swin and GLAM backbones, these networks are trained independently.

## 4.2 Results

We compare relay tokens against standard scaling baselines and recent state-of-the-art (SOTA) models in terms of precision and efficiency.

**Quantitative Analysis.** Table 1 demonstrates that relay tokens consistently outperforms the sliding-window (SW) baseline across all four evaluated datasets and four backbone architectures, achieving substantial improvements: up to +5.9 on Archaeoscape, +5.4 on URUR, and +15.9 on Gleason, and +8.0 mIoU on Cityscapes. An exception to this observation is the case of vanilla ViT on Archaeoscape, which have already been observed to perform poorly (Perron et al., 2024), and relay tokens do not appear to significantly improve this performance.

Despite being a simple extension of ViT-based architectures, relay tokens match or surpass specialised multi-scale methods such as ISDNet, GLNet, and SGNet across all UHR datasets. Transformer variants with linear attention, such as Flatten Swin, already perform competitively but are consistently improved by the addition of relay tokens. Linear-complexity Mamba models (Vision-Mamba and xT), while capable of processing full images in a single pass, generally underperform in our setting. CNN-based baselines remain computationally efficient but lag behind in accuracy.

**Comparison to SOTA.** Our main objective in this paper is not to chase incremental SOTA improvements against highly specialised approaches but to demonstrate how a simple, practical strategy can significantly enhance standard ViT-based architectures. Nevertheless, a SwinV2 backbone equipped with relay tokens achieves a clear SOTA on Archaeoscape by over 5 points, and trails by only 0.5 mIoU points behind the highly specialised remote sensing model WSDNet (Ji et al., 2023) on URUR without any structural modifications. On Gleason, public test servers are no longer available and recent works rely on private splits (Wang et al., 2024), which complicates fair comparison; evaluated on our clean train/test split, Flatten Swin+relay tokens outperform all other approaches.

Although Cityscapes uses higher-resolution images than most vision benchmarks, it is not an UHR dataset. We therefore do not expect relay tokens to reach state-of-the-art performance against highly specialised architectures such as SerNetFormer (Erişen, 2024), which reports 87.4 mIoU. Instead, our goal is to demonstrate that relay tokens provide a lightweight, efficient, and easily integrable improvement to standard vision transformer pipelines. Accordingly, we do not rely on practices commonly used to maximise performance on Cityscapes (Chen et al., 2018; Tao et al., 2020; Xie et al., 2021), such as pretraining on the 20K coarse annotations or on Mapillary Vistas (Neuhold et al., 2017), nor do we apply 10-scale test-time augmentation. Under these conditions, Flatten Swin and GLAM with relay tokens report the highest performance among all evaluated methods.

**Qualitative Analysis.** Figure 4 illustrates how relay tokens provide more coherent and contextually aware predictions across datasets. On Cityscapes, the local-only model mislabels partial objects (*e.g.*, identifying a cyclist's torso as a pedestrian), whereas including global context resolves such ambiguities. Likewise, on Archaeoscape, global windows help identify large water features (blue) and earth structures (yellow). On URUR, fields (light green) are better delineated against barelands (grey) and woodlands (dark green). For Gleason, the sliding window approach is not able to consistently classify the cell as malignant (blue), and predicts either normal tissue (white) or another cancer class (green).

**Comparison to Other Scaling Approaches.** We benchmark four common alternatives:

- **Sliding Window.** We independently segment local windows $x^{\text{local}}$ and stitch predictions, with an overlap ratio of 50% for Archaeoscape and 0% elsewhere.
- **Sliding Window + Registers.** We add relay tokens but process *only* the local window. This amounts to adding registers (Darcet et al., 2024).
- **Downsampling.** We process only the global window at a reduced resolution. If the image remains too large, we slide the global window.
- **Decision Fusion.** We process local and global windows in parallel, upsample the global prediction to the local resolution, and average the final logits.

Table 3: **Scaling Baselines**. We evaluate several scaling strategies with a SwinV2 backbone. We report the improvement with respect to the sliding window baseline with a red-to-green colour scale  -10  +0  +10 .

| Scaling strategy | | Archaeoscape | | URUR | | Gleason | | Cityscapes (val) | |
|---|---|---|---|---|---|---|---|---|---|
| Sliding window | SW: 256 | 51.9 | - | 41.0 | - | 48.1 | - | 68.2 | - |
| Registers | SW: 256 | 53.2 | +1.3 | 41.4 | +0.4 | 47.8 | -0.3 | 68.0 | -0.2 |
| Downsampling | SW: 1024↓4 | 47.1 | -4.8 | 45.3 | +4.3 | 55.1 | +7.0 | 61.8 | -6.4 |
| Decision fusion | MS: 256+1024↓ 4 | 56.5 | +4.6 | 43.8 | +2.8 | 55.1 | +7.0 | 72.3 | +4.1 |
| **Relay tokens (ours)** | MS: 256+1024↓ 4 | **57.8** | +5.9 | 45.6 | +4.6 | **57.0** | +8.9 | 75.1 | +6.9 |

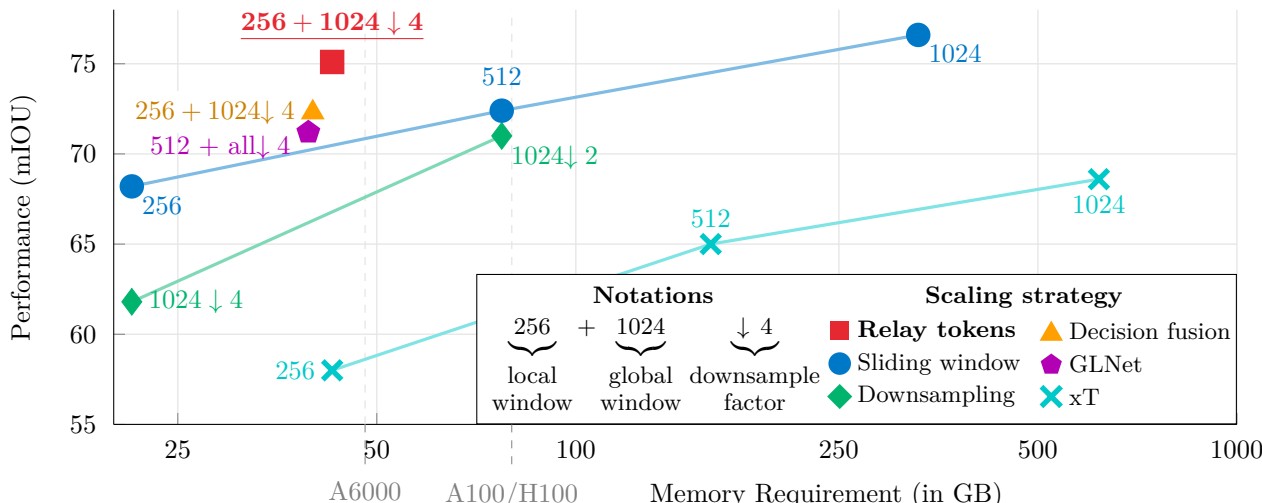

Figure 5: **Performance vs. Memory**. We plot the training GPU memory requirement and performance of a SwinV2 on Cityscapes for several scaling strategies. Relay tokens offer a better trade-off between memory and precision. In particular, we can train a SwinV2 model with relay tokens on a GPU with 48GB of RAM while outperforming models that require 8 80GB GPUs.

Table 3 shows that decision fusion is the strongest baseline, yet relay tokens outperform it by exchanging multi-scale features at every block rather than only at the output. Registers alone yield small, inconsistent gains, confirming that performance stems from genuine cross-scale interaction.

**Complexity Analysis.** Adding relay tokens to an existing ViT or Swin model introduces only a small number of additional learnable parameters: the relay tokens themselves and an additional copy of the projector. In practice, this results in less than 1.7% additional parameters for ViTs and 0.7% for Swin variants. Adding one relay token to a ViT-S adds only 384 parameters (0.002% of the model) and 96 for SwinV2-S (0.00014%); the remaining additional parameters come from the resolution-specific projectors.

In contrast, methods such as ISDNet (Guo et al., 2022) and GLNet (Chen et al., 2019) substantially increase their parameter count by using separate backbones for local and global processing, in addition to other auxiliary modules. Tab. 4 compares parameter counts and GFLOPs of different models.

Table 4: **Complexity Analysis.** For a 256 px input, without sharing the patch encoder.

| Model | #Param. [M] | GFLOPs |
|---|---|---|
| ViT | 23.9 | 6.6 |
| ViT + relay tokens | 24.3 (+1.6%) | 13.6 (+106%) |
| SwinV2 | 64.7 | 12.6 |
| SwinV2 + relay tokens | 65.1 (+0.9%) | 26.0 (+106%) |
| GLAM | 132 (+1.6%) | 75.4 |
| GLAM + relay tokens | 133 (+0.7%) | 152 (+102%) |
| DeepLabv3 | 39.6 | 41.0 |
| xT | 166 | 20.8 |

**Performance *vs*. Memory Trade-off.** Although processing dual-scale (local and global) windows approximately doubles the memory and computational costs compared to single-scale sliding-window baselines (see Tabs. 4 and 5), relay tokens remain significantly more efficient than approaches operating at high resolution. Figure 5 illustrates this efficiency-accuracy trade-off for Cityscapes: using relay tokens and a single 48 GB GPU, we achieve competitive performance to a SwinV2 applied with a $1024 \times 1024$ window, which requires eight 80 GB GPUs to train. Decision Fusion also presents a comparable efficiency trade-off but still falls short in accuracy.

Table 5: **Computational Costs.** Training time and throughput for a 256 px input.

| Backbone | Epoch time (minutes) | Throughput (full img/s) | |
|---|---|---|---|
| | Train | Train | Test |
| ViT | 32 | 9.3 | 31.1 |
| ViT + relays | 78 | 4.5 | 14.5 |
| SwinV2 | 33 | 3.1 | 9.9 |
| SwinV2 + relays | 75 | 1.4 | 4.7 |

**Test Set Performance for Cityscapes.** Due to limited Cityscapes evaluation server submissions, we test only two SwinV2-based configurations. The sliding-window baseline reaches 69.1% mIoU, while adding relay tokens improves it substantially to 75.0% (+5.9 points). These test set results closely align with our validation set outcomes, confirming the robustness of our experimental protocol.

### 4.3 Analysis

Despite its conceptual simplicity, our approach reveals valuable insights into how context and resolution interplay in modern segmentation architectures. Below, we investigate several critical aspects.

**How Large Should the Context Be?** The effectiveness of additional global context varies across datasets. For instance, Gleason images contain over 25M pixels each, strongly benefiting from extended spatial context. We represent in Fig. 6 the performance improvements relative to the sliding-window baseline across various spatial extents and downsampling ratios. We observe diminishing returns beyond a global context of 1024 pixels with a fixed downsampling ratio, especially given that it comes at a high cost due to quadratic growth of the attention matrix; for this reason, we were not able to to run the experiments with a $4096 \downarrow 4$ global window. Conversely, excessively coarse global windows (e.g., downsampling ratios beyond $8\times$ for a fixed global size) provide limited benefit. This highlights a dataset-dependent trade-off between memory, coverage, and resolution.

**How many Tokens Should One Use?** Figure 7 explores the impact of the token count. Performance gain saturates after 4 relay tokens, a common behaviour for register/global tokens (Zhang et al., 2021, Fig. 3). Computation grows quadratically at a moderate pace: only +26% from 1 to 32 tokens.

Figure 6: **Impact of Global Window Extent.** Performance as a function of the global window size in pixels at full image resolution. Two variants are tested: a fixed downsampling ratio ($\downarrow 4$) and a fixed global window size ($256 \times 256$).

| Datasets | Downsampling |
|---|---|
| Gleason | |
| Cityscapes | ——— Fixed $\downarrow 4$ |
| Archaeoscape | - - - - To $256 \times 256$ |
| URUR | |

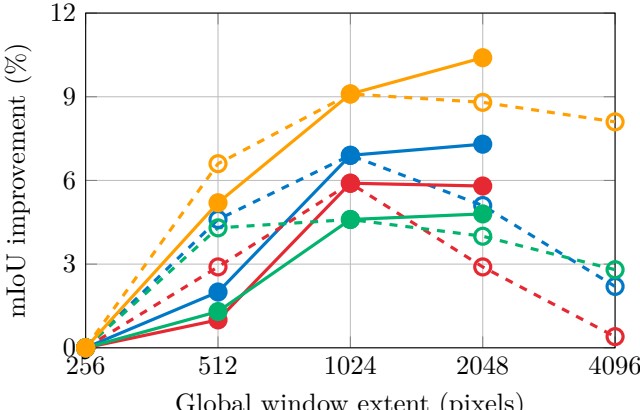

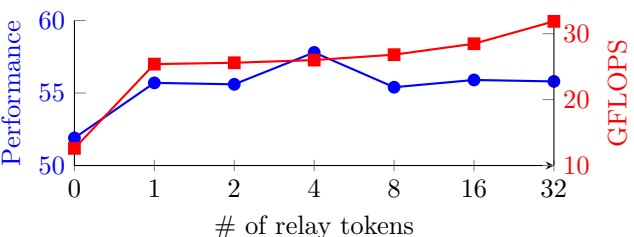

Figure 7: **Impact of Relay Count.** We evaluate several configurations for a SwinV2 model on Archaeoscape.

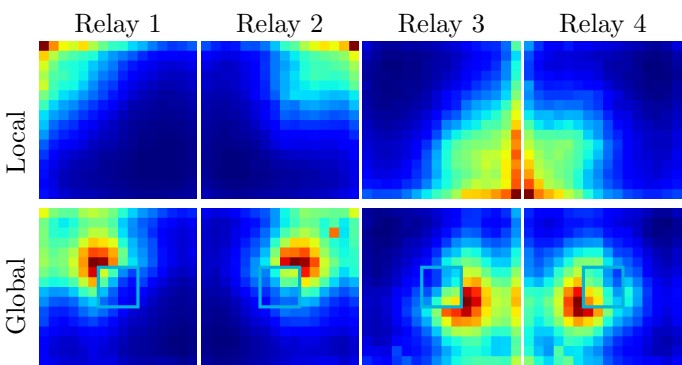

Figure 8: **Relay Tokens Attention Maps.** We compute the average attention between the relay tokens and each of the patch tokens of the local and global windows, averaged for all heads, blocks, and 320 random images of Cityscapes. Specialised, complementary spatial patterns emerge.

**Where Do Relay Tokens Look?** In Fig. 8, we visualise the average attention maps of four relay tokens in the local and global windows; more illustrations can be found in the Appendix. Two notable patterns emerge:

- *Radialisation:* Each relay token consistently specialises in a particular direction of the image (*e.g.*, NE, NW, SE, SW corners). This phenomenon persists across blocks and heads, presumably because token-wise MLP operations favour consistent specialisation.
- *Local–Global Harmonisation:* As a given relay token focuses in a corner of the local window, it also gathers global information in the surrounding area of the global window. This suggests that the networks processing each window are able to use the relay token to gather and propagate information in a spatially consistent manner. Moreover, the global attention avoids the centre of the image, where the local window already contains more fine-grained information.

**Which Objects Benefit Most From a Large Context?** We measure which object categories gain most from relay tokens compared to a sliding-window baseline (SW) using the relative IoU error improvement: $(\text{IoU}_{\text{relay}} - \text{IoU}_{\text{SW}})/(1 - \text{IoU}_{\text{SW}})$.

The classes of Cityscapes with the most gains are medium-scale objects (truck $+44\%$, bus $+39\%$, train $+33\%$, motorcycle $+25\%$, car $+20\%$) or classes defined by their function rather than appearance (road $+39\%$, sidewalk $+24\%$). Smaller objects with fine-grained structure (fence $+13\%$, pole $+6\%$, traffic light $+9\%$, and traffic sign $+8\%$) gain less from coarse global cues. Similar trends appear across datasets: ancient Khmer reservoirs in Archaeoscape ($+16\%$), greenhouses ($+15\%$) in URUR, and malignant cells ($+35\%$) in Gleason also notably benefit from the enhanced context.

### 4.4 Ablations

We evaluate the impact of different design choices on the performance and efficiency of our proposed approach.

**Architecture Ablation.** We report the impact on performance of several key architectural design choices in Tab. 6.

- **Losses.** Removing either the global loss $\mathcal{L}^{\text{global}}$ or the consistency loss $\mathcal{L}^{\text{cons}}$ degrades performance by approximately 1.0 to 1.9 mIoU points, respectively. Simultaneously omitting both losses leads to a larger performance drop of 2.5 points, demonstrating their individual yet complementary contributions.

Table 6: **Impact on Performance.** Effect of individual design choices evaluated in two representative settings: `conf.A` — ViT on Gleason, and `conf.B` — SwinV2 on Archaeoscape. Grey denotes unchanged configuration from the default .

| | A | B | | A | B |
|---|---|---|---|---|---|
| Relay Tokens | **49.1** | **57.8** | 2 relays | 49.3 | 51.2 |
| Sliding Window | 33.2 | 51.9 | 8 relays | 46.3 | 55.7 |
| no $\mathcal{L}^{\text{cons}}$ | 47.2 | 52.1 | Token Concatenation | 46.8 | - |
| no $\mathcal{L}^{\text{global}}$ | 47.8 | 50.9 | Shared Projectors | 49.1 | 49.6 |
| no $\mathcal{L}^{\text{cons}}$ no $\mathcal{L}^{\text{global}}$ | 48.1 | 48.6 | Different Projectors | 46.3 | 57.8 |

Table 7: **Impact on Efficiency.** We evaluate on the Cityscapes validation set the performance and efficiency of different relay-token variants: Fewer Blocks match the baseline compute, and Parallel Relays facilitate parallelisation.

| Model | Variant | mIOU | FLOPs | Throughput (img/s) |
|---|---|---|---|---|
| ViT | Full | 53.0 | 6.6 | 995 |
| ViT + Relay | Full | 61.0 | 13.6 | 465 |
| ViT + Relay | Fewer Blocks | 55.0 | 6.8 | 898 |
| ViT + Relay | Parallel Relays | 56.4 | 13.6 | 488 |

- **Number of Relay Tokens.** We vary the number of relay tokens, testing configurations with 2, 4 (default), and 8 tokens. Using only 2 relay tokens yields insufficient improvements over the baseline, while employing 8 relay tokens is slightly less effective than 4. This finding aligns with observations on register token parameterisation for segmentation tasks (Darcet et al., 2024, Fig. 8). Similar trends occur across other datasets, although results might differ for datasets with more intricate multi-scale interactions.
- **Token Concatenation.** An alternative multi-scale approach is to concatenate tokens from local and global windows before processing them through a single network (without relay tokens). While straightforward for plain ViT architectures, this approach is complex for Swin models due to their window-based attention mechanism. It also significantly increases the computational footprint (approximately four times the baseline, twice that of relay tokens). Despite this additional resource requirement, concatenation performs worse than relay tokens, suggesting that relay tokens are more effective to capture multi-scale features than full self-attention between patches.
- **Shared Projectors.** The networks processing the local and global windows share over 99% of their weights, except the relay token and in some cases, the projectors. Whether or not sharing projectors is beneficial to the model is very dataset specific: on Gleason using different projectors causes a loss of -2.8% (ViT) while it improves the performance by +8.2% on Archaeoscape (SwinV2). This underlines how different datasets can have widely different scale-invariance properties, which can justify (in the case of Gleason and URUR) a shared processing across scales.
- **Adaptive Positional Encoding.** Inspired by ScaleMAE (Reed et al., 2023), we explore using positional encoding whose scale depends on the downscaling factor. Contrary to expectations, this approach noticeably degraded performance. We hypothesise this is due to incompatibility with positional encodings learned during the model's initial pretraining.

**Relay Token Variants.** By design, introducing relay tokens approximately doubles the number of transformer blocks and the computational load. To ascertain whether the gains come from additional compute or from improved cross-scale interaction, we run two targeted experiments in Tab. 7:

- **Fewer Blocks.** We keep only the first six out of twelve transformer blocks in both the local and global branches, thus matching the computational cost of the baseline model (ViT-S with sliding window). This variant still perform better than the ViT baseline, indicating that relay tokens provide benefits beyond pure compute scaling. While it underperforms the full model, this variant offers a strong accuracy–efficiency trade-off in compute-constrained settings.
- **Parallel Relay.** A limitation of our default design is that global and local windows are processed sequentially, which reduces parallelism. We therefore consider a *parallel relay* variant in which global and local branches are executed concurrently, each with its own copy of the relay tokens. The updated relay representation is obtained by averaging the tokens produced by both branches. This design improves parallelization relative to the sequential *global–to–local* scheme, at the cost of weaker cross-scale coupling. Specifically, we replace Eqs. (4) and (5) in the paper by:

$$\left[f_{b+\frac{1}{2}}^{\mathrm{relay(global)}}, f_{b+1}^{\mathrm{global}}\right] = \phi_b^{\mathrm{block}}\left([f_b^{\mathrm{relay}}, f_b^{\mathrm{global}}]\right) \tag{11}$$

$$\left[f_{b+\frac{1}{2}}^{\mathrm{relay(local)}}, f_{b+1}^{\mathrm{local}}\right] = \phi_b^{\mathrm{block}}\left([f_b^{\mathrm{relay}}, f_b^{\mathrm{local}}]\right) \tag{12}$$

$$f_{b+1}^{\mathrm{relay}} = 0.5(f_{b+\frac{1}{2}}^{\mathrm{relay(global)}} + f_{b+\frac{1}{2}}^{\mathrm{relay(local)}}) \tag{13}$$

This yields only modest throughput gains, suggesting that GPU utilisation is already efficient with the the original sequential relay. However, the performance improvement relative to the ViT baseline drops significantly, suggesting that sequential relay feedback is important for delivering effective, scale-adaptive feedback.

## Conclusion

We introduced relay tokens, a lightweight extension of vision transformer models, enabling effective multi-scale reasoning for large-image segmentation. With minimal additional parameters and straightforward implementation, relay tokens significantly boost segmentation accuracy across diverse datasets and domains. Our method offers an efficient, easily deployable alternative to specialised multi-scale or large-scale segmentation architectures, enabling practitioners to seamlessly integrate multi-scale reasoning capabilities into existing transformer-based workflows.

## Acknowledgements and disclosure of funding

The experiments conducted in this study were performed using HPC/AI resources provided by GENCI-IDRIS (Grants 2024-AD011014781R1, 2024-AD011015196R1, 2025-AD011014713R2).

This work was supported by the ANR project sharp ANR-23-PEIA-0008 in the context of the PEPR IA, by Hi! PARIS under the France 2030 program (ANR-23-IACL-0005) and by Paris Ile-de-France Region through the DIM PAMIR network (IDF-DIM-PAMIR-2025-4-021). We thank Nicolas Dufour, Antoine Guédon, and Lucas Ventura for inspiring discussions and valuable feedback.

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

## A-1 Implementation

We provide in Algorithm A-1 an extended implementation of a ViT with relay tokens.

---

Algorithm A-1: **ViT with Relay pseudocode.**

---

(a) ViT with Relay. Minimal implementation.

```python
class ViT_relay(nn.Module):
  def __init__(self, B, D, P, in_C, out_C, R):
    super().__init__()
    self.P = P
    self.proj = nn.Linear(P * P * in_C, D)
    self.blocks = nn.ModuleList(
      [Transformer(D) for _ in range(B)])
    self.out_head = nn.Linear(D, P * P * out_C)
    self.relay = nn.Parameter(torch.randn(R, D))

  def forward(self, img_loc, img_glo):
    BS = img_loc.size(0) # batch size

    p_loc = patchify(img_loc, self.P)
    r_glo = patchify(img_glo, self.P)

    f_loc = self.proj(p_loc) + pos_enc(P, D, BS)
    f_glo = self.proj(r_glo) + pos_enc(P, D, BS)

    f_r = repeat(self.relay, BS)

    for block in self.blocks:
      f_r_glo = block(torch.cat((f_r, f_glo),1))
      f_glo, f_r = f_r_glo[:, R:], f_r_glo[:, :R]

      f_r_loc = block(torch.cat((f_r, f_loc),1))
      f_loc, f_r = f_r_loc[:, R:], f_r_loc[:, :R]

    out_patches = self.out_head(f_loc)
    out = unpatchify(out_patches, self.P)
    return out
```

(b) ViT with Relay. Auxiliary functions.

```python
import torch
import torch.nn as nn
from einops import rearrange

def patchify(img, patch_size):
  # (B, C, H, W)
  # -> (B, #patches, patch_size^2 * C)
  return rearrange(
    img,
    'b c (hp p) (wp q) -> b (hp wp) (c p q)',
    p=patch_size, q=patch_size
  )

def unpatchify(patches, patch_size, H, W):
  # (B, #patches, patch_size^2 * out_channels)
  # -> (B, out_chan, H, W)
  return rearrange(
    patches,
    'b (hp wp) (oc p q) -> b oc (hp p) (wp q)',
    hp=H // patch_size, wp=W // patch_size,
    p=patch_size, q=patch_size
  )

class TransformerBlock(nn.Module):
  def __init__(self, dim):
    super().__init__()
    self.attn = nn.MultiheadAttention(dim,
      num_heads=H, batch_first=True)
    self.mlp = nn.Sequential(
      nn.Linear(dim, 4 * dim),
      nn.ReLU(),
      nn.Linear(4 * dim, dim)
    )

  def forward(self, x):
    x = x + self.attn(x, x, x)[0]
    x = x + self.mlp(x)
    return x

def repeat(reg, batch_size):
  # Replicate relay tokens across
  #the batch dimension:
  return reg[None].expand(batch_size, -1, -1)
```

## A-2    Additional Analysis

**Attention Map**    In Fig. A-1, we present average attention maps of ViT trained on Cityscapes with 2, 4 and 8 relay tokens. For all configurations, we observe strong coordination between scales: each relay token focuses on a well-defined part of the local window and the surrounding area of the global window. The radialisation observed with 4 relay tokens mostly holds with most token attending to a single cardinal direction of the window. Interestingly, one of the 8 relay tokens attends the centre part of both the local and global window. This may allow the centre of the local window to benefits from long distance contextual information contain in the global window.

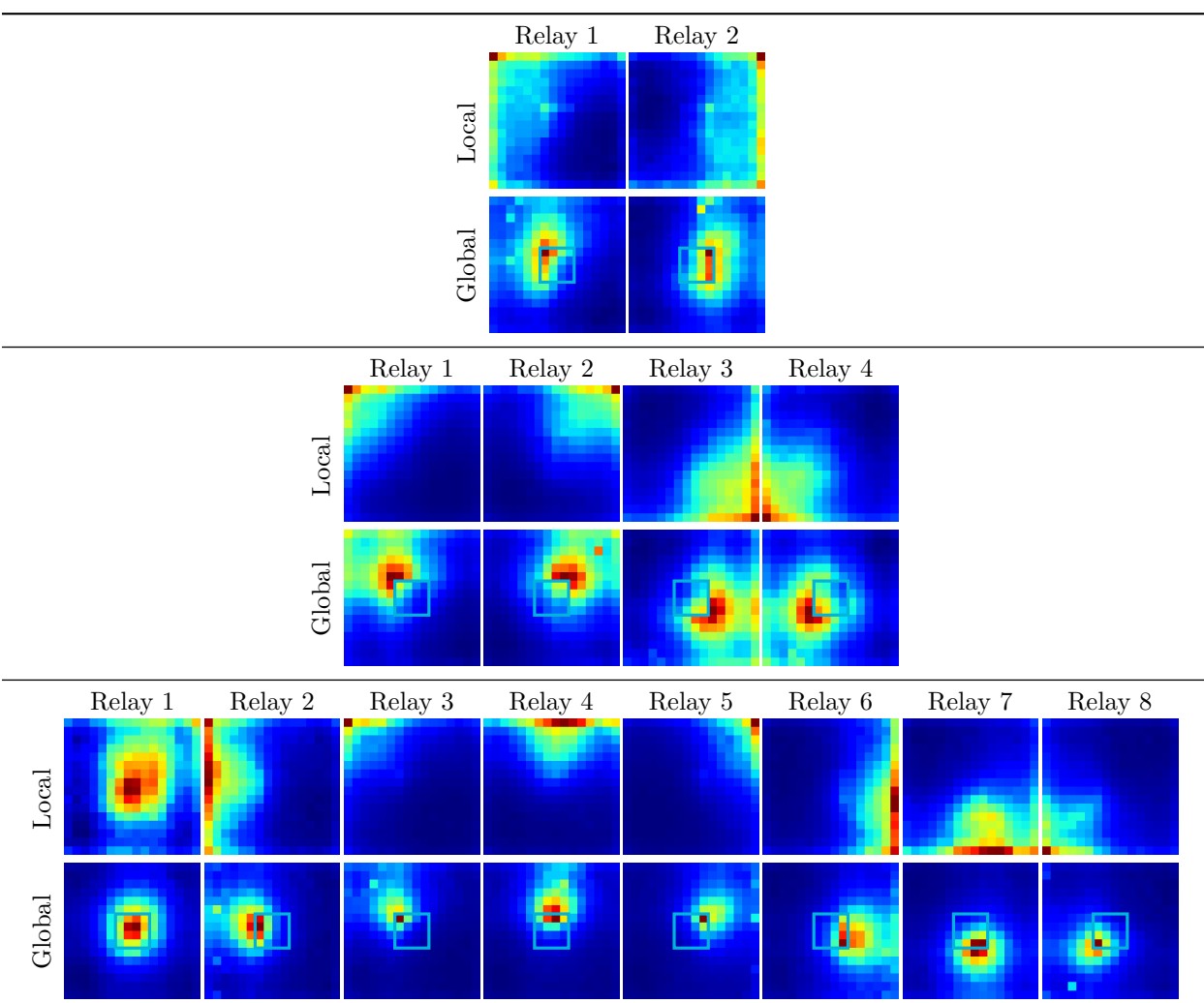

Figure A-1: **Attention.** Attention maps of the relay tokens for a ViT trained on Cityscapes with 2, 4, 8 relay tokens.

**Model Size**    In Tab. A-1, we provide the performance of our approach with different SwinV2 backbone sizes on the Cityscapes dataset. We observe that our approach with relay token show similar scaling to the standard SwinV2 model.

**FLatten Transformer performance**    In Tab. A-2 we compare the memory usage and throughput of the Swin configuration of FLatten Transformer (Han et al., 2023) on Cityscapes (batch size = 64). We observe that adding relays yields higher performance gains with a lower memory footprint compared to

| Model | Tiny | Small | Base | Large |
|---|---|---|---|---|
| SwinV2 | 64.9 | 68.2 | 68.1 | 67.0 |
| SwinV2 + relay token | 73.5 | 75.1 | 75.1 | 74.1 |

Table A-1: **Impact of model size on performance on Cityscapes.**

| Model | mIoU | GPU memory (GB) | Throughput (full img/s) |
|---|---|---|---|
| Flatten Swin-S / 256 | 68.0 | 26 | 10.6 |
| Flatten Swin-S / 512 | 73.5 | 90 | 9.5 |
| Flatten Swin-S / 256 + RELAY | 75.3 | 48 | 4.9 |

Table A-2: **Memory Usage and Throughput of Flatten Swin on Cityscapes.**

simply increasing the sliding-window size, consistent with our observations in Fig. 5 of the main paper. While linear-attention models can scale to larger windows with limited computational overhead and thus remain faster than relay-based variants, they incur substantially higher memory costs in our experiments.

## A-3 Datasets

We provide in Fig. A-2 the colour codes for the classes of our evaluated datasets.

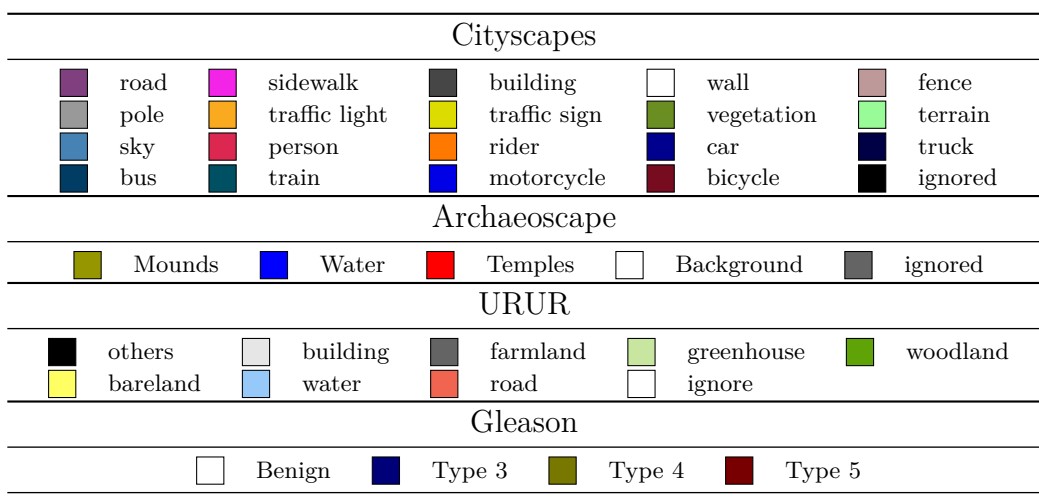

Figure A-2: **Colourmaps.** We provide the colour code for the classes of all the dataset evaluated.

