# OpenReview forum: "Adapting Vision Transformers to Ultra-High Resolution Semantic Segmentation with Relay Tokens"
_TMLR — Accepted by TMLR_

### Review · Reviewer_EHcX · 2025-09-17

**Summary Of Contributions:**

This paper introduces a simple and effective method for segmenting ultra-high-resolution images for vision transformers.
It simultaneously processes the image by the local and global branches, with the introduced relay tokens to preserve local details and global awareness.
Experiments demonstrate that the proposed method improves the performance of multiple vision transformers with only a small number of additional parameters.

strengths:
1. The writing is clear and easy to follow.
2. The proposed method is simple and effective, and it is easy to implement for various vision transformers.

weaknesses:
1. Although the additional parameters are small, the additional computational cost is huge.
According to Table 4, the GFLOPs will be doubled after applying the method to vision transformers.
This may limit the practical applications of the proposed method.
2. The experiments only apply the method to some basic vision transformers, such as ViT and Swin.
It would be more convincing if the authors could apply the method to some advanced vision transformers, such as GPWFormer [1] and BPT [2].
If not, the authors should provide some discussion on why the method is not suitable for advanced vision transformers.
3. The introduction is too short. To make the paper more self-contained and friendly to readers, the authors should expand the introduction section by providing more background information and related works.
4. The paper mentioned "The quadratic token–token interaction in vanilla transformer layers renders off-the-shelf ViT-like models infeasible for UHR semantic segmentation".
However, there are some efficient attention mechanisms that can reduce the computational cost of attention, such as Linformer [3], Efficient Attention [4], etc.
The authors should discuss the advantages of the proposed method compared to these efficient attention mechanisms, and provide some experimental results if possible.

[1] Ji, Deyi, Feng Zhao, and Hongtao Lu. "Guided patch-grouping wavelet transformer with spatial congruence for ultra-high resolution segmentation." Proceedings of the Thirty-Second International Joint Conference on Artificial Intelligence. 2023.

[2] Sun, Haopeng, et al. "Ultra-high resolution segmentation via boundary-enhanced patch-merging transformer." Proceedings of the AAAI Conference on Artificial Intelligence. Vol. 39. No. 7. 2025.

[3] Wang, Sinong, et al. "Linformer: Self-attention with linear complexity." arXiv preprint arXiv:2006.04768 (2020).

[4] Shen, Zhuoran, et al. "Efficient attention: Attention with linear complexities." Proceedings of the IEEE/CVF winter conference on applications of computer vision. 2021.

**Audience:**

Yes

**Audience Explanation:**

Ultra-high resolution semantic segmentation is a practical problem, and researchers working on this topic will be interested in this paper.

**Claims And Evidence:**

Yes

**Claims Explanation:**

The author evaluates the proposed method and baselines on multiple datasets, with ablation studies to validate the effectiveness of each component.

**Requested Changes:**

Please see the weaknesses.

---

> ### Author Response · Authors · 2025-10-20
> **Thank your for the many suggestions!**
>
> We thank the reviewer for their valuable comments and suggestions, which we have answered below.
>
> **Computational efficiency:**
> We refer the reviewer to the new experiments presented in our response to *Reviewer k6WA*, where we explicitly analyze the impact of relay exchanges under reduced computation budgets (fewer blocks and parallel relay connections). These results clarify that the performance gains stem from the proposed cross-scale relay mechanism rather than from increased computational cost alone.
>
> **Introduction:** We agree and will expand the introduction to provide additional background and related work, particularly on multi-scale feature aggregation and efficient attention mechanisms.
>
> **Additional backbones and linear attention.**
> We appreciate the suggestion to include more advanced transformer architectures such as GPWFormer and BPT. Unfortunately, these models do not currently release their implementations, preventing us from integrating relay tokens. Nevertheless, since our method is designed as a drop-in module, we expect it to generalize seamlessly to such backbones once public code becomes available. However, we were able to consider an additional backbone below
>
> Following the reviewer’s request to consider vision transformers with linear-attention schemes, we added relays to FlattenTransformer [1], an efficient architecture employing linear attention (similar to LongFormer). Using its Swin-based configuration, we observe comparable relative gains when adding relay tokens, confirming that our mechanism complements efficiency-oriented models as well. We summarize preliminary results on Archaeoscape, URUR, Gleason and Cityscapes below. Full experiments across all benchmarks will be included in the camera-ready version.
>
> | Model              | Scaling strategy (px) | Archaeoscape |     URUR   |  Gleason | Cityscapes |
> |--------------------|:---------------------:|:------------:|:----------:|:--------:|:----------:|
> | FL-Swin_S          |                   256 |         53.4 |     43.0   |     50.7 |       70.2 |
> | FL-Swin_S          |                   512 |         52.9 |     45.1   |     54.0 |       73.6 |
> | FL-Swin_S + relays |          256 + 1024÷4 |     **55.2** | **45.8**   | **57.7** |   **77.5** |
>
> **Memory and Computation:** We also compare the memory usage and throughput of this model on Cityscapes (batch size = 64) in the table below. We observe that adding relays yields higher performance gains with a lower memory footprint compared to simply increasing the sliding-window size, consistent with our observations in Figure 4 of the main paper. While linear-attention models can scale to larger windows with limited computational overhead and thus remain faster than relay-based variants, they incur substantially higher memory costs in our experiments.
>
> |                       |     mIoU | GPU mem (GB) | Throughput (full img/s) |
> |-----------------------|:----------:|:--------------:|:-------------------------:|
> | FL-Swin_S/256         |     68.0 |           **26** |                    **10.6**|
> | FL-Swin_S/512         |     73.5 |           90 |                     9.5 |
> | FL-Swin_S/256 + relays | **75.3** |           48 |                     4.9 |
>
> [1] Dongchen Han, Xuran Pan, Yizeng Han, Shiji Song, and Gao Huang. "FLatten Transformer: Vision Transformer using Focused Linear Attention". In ICCV, 2023

---

### Review · Reviewer_Drrm · 2025-10-03

**Summary Of Contributions:**

This paper proposes relay token as method to improve the semantic segmentation of ultra-high-resolution images. Relay token uses a global branch that works on a down-sampled image patch to provide global information and a fine-grained local branch that works on local image patch. The relay tokens are shared between the global and local branches to transfer information between them. Relay token is general across vision Transformer models and requires to add only a small number of parameters. Experiment results show that relay token improves the accuracy of the semantic segmentation.

**Audience:**

Yes

**Audience Explanation:**

This paper proposes a simple, lightweight yet general and strong method for the semantic segmentation of ultra-high-resolution images. Many audiences of TMLR will be interested to learn it.

**Claims And Evidence:**

Yes

**Claims Explanation:**

I really like this paper due to its succinct presentation, simple idea, good generality, and extensive experiments. The strong points of the paper include

S1: Existing semantic segmentation methods for ultra-high-resolution images are discussed comprehensively along with their limitations (i.e., Figure 1).

S2: The relay token method is simple and general across different vision Transformer model.

S3: The experiment results are strong, showing that relay token effectively improves the accuracy of semantic segmentation. Other details, such as the resolution of the two branches, the number of relay tokens, and the designs, are also evaluated.

S4: Relay token is compared with other methods that adopt the global-local idea, and reasons are provided why the relay token is a better choice.

**Requested Changes:**

R1: Please add some efficiency experiments regarding (i) the training time of relay token and (ii) the inference speed with relay tokens.

---

> ### Author Response · Authors · 2025-10-20
> **Training and inference speed**
>
> We thank the reviewer for requesting the training and inference times, as these are key figures for assessing efficiency. The table below reports the training time per epoch and inference throughput on Cityscapes for ViT and Swin backbones, both with and without relay tokens. All runs use the same optimizer, data augmentations, mixed precision, and training schedule. Inference is measured at matched input resolution, batch size, and kernel settings.
>
> | Backbone      |   Train time (min per epoch) |  Throughput: Train (full img/s) | Throughput: Test |
> |---------------|:------------------:|:----------------:|:------:|
> | ViT           |               32 |            9.3 | 31.1 |
> | ViT + relays  |               78 |            4.5 | 14.5 |
> |    |   |   |  |
> | Swin          |               33 |            3.1 |  9.9 |
> | Swin + relays |               75 |            1.4 |  4.7 |
>
> **Analysis:**
> As expected, introducing relay tokens increases training time and reduces single-stream throughput, yet consistently improves accuracy at comparable parameter counts. Note that training time is largely dominated by data loading, which is why we also report the throughput of the model alone. We encourage the reviewer to consider our additional experiments in our comment to *Reviewer k6WA*, where we disantangle the performance impact of additional computation and cross-scale communication.

---

### Review · Reviewer_k6WA · 2025-10-12

**Summary Of Contributions:**

The paper tackles the challenge of ultra-high-resolution semantic segmentation by introducing relay tokens - a small set of learnable query tokens shared between local and global transformer branches. The local transformer blocks process patches from the high-resolution image, while the global transformer blocks operate on patches from a downsampled version of the same image.
The method interleaves global and local transformer blocks such that the relay tokens are first updated by the global block and then passed to the local block, allowing efficient information exchange across scales.

The paper demonstrates competitive performance across multiple segmentation benchmarks, supported by both quantitative and qualitative results. The ablation studies are thorough, analyzing the effect of the number of relay tokens, the loss components, and the global window size. The visualization of average attention maps further confirms that different relay tokens attend to spatially distinct image regions, as expected.

The main limitation of the paper is computational cost: the proposed approach roughly doubles the FLOPs compared to single-scale baselines, since it requires running both local and global transformer branches (albeit with shared weights). The contribution would be stronger if the paper had investigated whether similar performance could be achieved with reduced compute - for example, by reducing the number of transformer layers or by using relay tokens less frequently.

**Audience:**

Yes

**Audience Explanation:**

This paper would appeal to TMLR readers interested in vision transformer architectures, multi-scale modeling, and efficient segmentation for large images. The proposed method is practical, general, and clearly positioned within ongoing research on scalable, context-aware transformer designs. Its insights are relevant to both researchers developing new architectures and practitioners working with ultra-high-resolution visual data.

**Broader Impact Concerns:**

This work focuses on a technical contribution to vision transformer efficiency and scalability, without direct societal or ethical implications. The proposed relay-token mechanism is a general modeling improvement and does not involve sensitive data, human subjects, or method-specific considerations.

**Claims And Evidence:**

Yes

**Claims Explanation:**

The claims are well supported by experimental results and ablations. The proposed relay-token mechanism is evaluated on Cityscapes, Archaeoscape, URUR, and Gleason, and compared against a wide range of baselines including ViT, Swin Transformer, Vision Mamba, and several specialized multi-scale architectures. The paper also contrasts different scaling strategies - sliding windows, registers, downsampling, and decision fusion (the closest baseline, which fuses local and global predictions via averaging) - and reports both accuracy and peak GPU memory usage.

Relay tokens increase the parameter count by only ~2%, while providing consistent performance gains. The main trade-off lies in higher computational cost (approximately 2× FLOPs). The qualitative attention maps are intuitive, showing that relay tokens specialize in distinct spatial regions, and the ablations convincingly demonstrate the importance of the loss terms, the number of relay tokens, and insight into whether local/global projectors should be shared.

**Requested Changes:**

The paper is well written and provides a clear discussion of the proposed method, its performance, computational cost (in terms of memory and FLOPs), and the intuition behind how relay tokens facilitate cross-scale information exchange. However, one clear drawback is that the approach roughly doubles the FLOPs compared to single-scale baselines and is inherently sequential, since it requires interleaving global and local transformer blocks.

Critical change for acceptance:
The authors should evaluate whether relay tokens can enable similar segmentation performance with reduced computation, for instance by using fewer transformer blocks or by performing relay exchanges only in a subset of layers to allow parallelization of local and global blocks. Such an ablation would clarify whether the observed improvements arise from the relay-token mechanism itself or simply from increased computation. Currently, most baselines (except the decision fusion method) use fewer FLOPs than the proposed approach, which makes it harder to contextualize the contribution despite comparable parameter counts.

Recommended change for clarity:
- Add a figure illustrating the spatial relationship between the cropped global predictions and the local features used in the losses in Equations (9) and (10). This would make the training setup and the alignment between global and local supervision much easier for readers to follow.

Minor edits (non-critical):
- Figure 3: Increase vertical spacing between the “Relay tokens” and “Sliding win.” rows to avoid overlapping text.
- Figure 7 and Table 5: Clearly specify the metric being reported as performance (e.g., mIoU) in captions or labels.

---

> ### Author Response · Authors · 2025-10-20
> **Thank you for insightful comments, which led to interesting experiments**
>
> ## Distangling the impact of computation and design
>
> We agree that, by design, introducing relay tokens approximately doubles the number of transformer blocks and the computational load. This makes it difficult to fully disentangle whether the observed gains stem from additional compute or from improved cross-scale interactions. Following the reviewer’s recommendation, we conducted two targeted ablations to isolate these effects
>
> 1) *Fewer Blocks*: We retain only the first six out of twelve transformer blocks in both the local and global branches, yielding the same total number of blocks (and approximately the same FLOPs) as a vanilla ViT-S.
>
> 2) *Parallel Relay*: We also agree that the sequential global → local design hinders parallelization. To address this, we evaluate an alternative parallel relay scheme in which both local and global branches operate simultaneously with duplicated relay tokens. The updated relay token is computed as the mean of the two updated tokens. Specifically, we replace Eqs. (4–5) in the paper by:
> \begin{align}
> [f_{b+1}^{\text{relay, local}}, f_{b+1}^{\text{local}}] =  \phi_b^{\text{block}}([f_b^{\text{relay}}, f_b^{\text{local}}]] )
> \end{align}
> \begin{align}
> [f_{b+1}^{\text{relay, global}}, f_{b+1}^{\text{global}}] =  \phi_b^{\text{block}}([f_b^{\text{relay}}, f_b^{\text{global}}]] )
> \end{align}
> \begin{align}
> f_{b+1} ^{\text{relay}}= \frac12(f_{b+1}^{\text{relay, local}}+f_{b+1}^{\text{relay, global}})
> \end{align}
>
>
> **Results:** We summarize preliminary results below (Cityscapes validation set):
>
> | Model       | Variant         | mIoU | FLOPs | Throughput (full img/s) |
> |-------------|:-----------------:|:------:|:-------:|:--------------------:|
> | ViT         | Full            | 53.0 |   **6.6** |                **31.1** |
> | ViT + Relay | Full            | **61.0** |  13.6 |                14.5 |
> | ViT + Relay | Fewer Blocks    | 55.0 |   6.8 |                28.1 |
> | ViT + Relay | Parallel Relays | 56.4 |  13.6 |                15.2 |
>
>
> **Analysis:** The *Fewer Blocks* variant, which maintains identical depth and compute as the baseline ViT but distributes computation across local and global streams, already outperforms the vanilla ViT. This confirms that the relay mechanism contributes to improved representation beyond simple compute scaling. While performance is below the full model, this variant offers a favorable trade-off for compute-constrained settings.
>
> The *Parallel Relay* variant allows simultaneous execution of both branches, leading to modest throughput gains. This suggests that GPU utilization is already efficient even with the original sequential updates for relays. However, the performance improvement relative to the ViT baseline drops significantly. We hypothesize that the parallel nature of the update prevents the relays from delivering effective, scale-adaptive feedback.
>
> We thank the reviewer again for this insightful suggestion, which prompted two interesting experiments. We hope this addresses your questions and requests regarding efficiency, and we would be happy to answer any follow-up inquiries.
>
> ## Additional Figure
> We will add a figure clarifying the spatial relationship between the cropped global and local predictions and features.

---

### Decision · Action_Editor_ZE6N · 2025-12-06

**Recommendation:** Accept as is

**Audience:**

Yes

**Audience Explanation:**

The paper will interest TMLR readers working on multi-scale modeling with vision transformers, and semantic segmentation of high-resolution images, which are practical and exciting topics.

**Claims And Evidence:**

Yes

**Claims Explanation:**

All reviewers agree that the paper’s claims are well supported by extensive experiments across multiple datasets, fair comparisons to baselines, and thorough ablations.